# Obtaining Specific Sequence Tags for *Yersinia pestis* and Visually Detecting Them Using the CRISPR-Cas12a System

**DOI:** 10.3390/pathogens10050562

**Published:** 2021-05-06

**Authors:** Gang Chen, Yufei Lyu, Dongshu Wang, Li Zhu, Shiyang Cao, Chao Pan, Erling Feng, Weicai Zhang, Xiankai Liu, Yujun Cui, Hengliang Wang

**Affiliations:** 1State Key Laboratory of Pathogens and Biosecurity, Beijing Institute of Biotechnology, 20 Dongdajie Street, Fengtai District, Beijng 100071, China; tony_chan2008@sina.com (G.C.); flygogo.cool@163.com (Y.L.); wangdongshu@foxmail.com (D.W.); jewly54@126.com (L.Z.); panchaosunny@163.com (C.P.); fengel@sohu.com (E.F.); ammsmelab@126.com (W.Z.); 2State Key Laboratory of Pathogen and Biosecurity, Beijing Institute of Microbiology and Epidemiology, 20 Dongdajie Street, Fengtai District, Beijing 100071, China; sycao90@sina.com

**Keywords:** *Yersinia pestis*, specific tags, CRISPR-Cas12a, visual detection, species discrimination

## Abstract

Three worldwide historical plague pandemics resulted in millions of deaths. *Yersinia pestis*, the etiologic agent of plague, is also a potential bioterrorist weapon. Simple, rapid, and specific detection of *Y. pestis* is important to prevent and control plague. However, the high similarity between *Y. pestis* and its sister species within the same genus makes detection work problematic. Here, the genome sequence from the *Y. pestis* CO92 strain was electronically separated into millions of fragments. These fragments were analyzed and compared with the genome sequences of 539 *Y. pestis* strains and 572 strains of 20 species within the *Yersinia* genus. Altogether, 97 *Y. pestis*-specific tags containing two or more single nucleotide polymorphism sites were screened out. These 97 tags efficiently distinguished *Y. pestis* from all other closely related species. We chose four of these tags to design a Cas12a-based detection system. PCR–fluorescence methodology was used to test the specificity of these tags, and the results showed that the fluorescence intensity produced by *Y. pestis* was significantly higher than that of non-*Y. pestis* (*p* < 0.0001). We then employed recombinase polymerase amplification and lateral flow dipsticks to visualize the results. Our newly developed plasmid-independent, species-specific library of tags completely and effectively screened chromosomal sequences. The detection limit of our four-tag Cas12a system reached picogram levels.

## 1. Introduction

Plague is a fatal infectious disease caused by *Yersinia pestis*. Plague pandemics, of which there have been three from the sixth century AD to the end of the 19th century, have inflicted a heavy toll on human societies. *Y. pestis* mainly infects humans through flea bites or inhalation and, without antibiotic treatments, patients die within a few days. *Y. pestis* is also a concern as a potential bioterrorist weapon [1].

*Y. pestis*, *Y. pseudotuberculosis*, and *Y. enterocolitica* are the main pathogenic bacteria in the *Yersinia* genus. The reference genome of the *Y. pestis* CO92 strain comprises a 4.65 Mb chromosome and three plasmids of 96.2 kb, 70.3 kb, and 9.6 kb [2]. The *Y. pestis* and *Y. pseudotuberculosis* genomes are very similar [3]. The *Y. pseudotuberculosis* IP32953 reference genome strain encodes 3974 predicted genes. Among them, 2976 genes share sequence homology levels of at least 97% with *Y. pestis* CO92 [3]. Sequence analysis of five housekeeping genes and one gene involved in lipopolysaccharide synthesis revealed that these *Y. pestis* genes are the same or almost the same as those from *Y. pseudotuberculosis*. In fact, the similarity is so high that some researchers have inferred that *Y. pestis* has evolved from *Y. pseudotuberculosis* [4]. Consistent with this observation, *Y. pestis* and *Y. pseudotuberculosis* 16S rRNA molecules are identical and therefore cannot be used to differentiate these species [5]. This presents great challenges for species identification of *Y. pestis*. With the increasing number of sequenced strains, the *Yersinia* bacterial genome database at the National Center for Biotechnology Information (NCBI) has become large and complicated. However, this database contains some non-*Y. pestis* genome data. Such data may have been deposited from simple sequence comparisons and may therefore mislead researchers in subsequent analyses. Thus, it is necessary to establish a simple and efficient method to accurately identify *Y. pestis*.

Currently, bacteriological, immunological, and molecular biological methods are commonly used to identify *Y. pestis*, and bacterial culture remains common. Medium containing cefsulodin–irgasan–novobiocin (CIN) is currently used as a selective medium for *Y. pestis* [6], but suspicious colonies on CIN medium take two days of cultivation at 28 °C for identification. Hence, this method is time-consuming and unsuitable for rapid diagnosis. The *Y. pestis* F1 antigen is the most commonly used immunological target, and many serological methods can be used to detect anti-F1 antibodies (e.g., enzyme linked immunosorbent assays, indirect hemagglutination assays, and gold-immunochromatography assays). Immunological detection methods based on the *Y. pestis* F1 antigen are completable within 15 min [7]. However, these methods lack sensitivity, and false negatives are not uncommon when testing sputum specimens. Earlier targets of the PCR-based methods used to identify *Y. pestis* were usually sequences on unique plasmids, but these targets can be ineffective at identifying atypical strains such as plasmid-deficient ones [8,9,10]. Two unique plasmids (pPCP1 and pMT1) were isolated during the evolution of *Y. pestis*, and a virulence plasmid (pCD1 of *Y. pestis*) is found in all pathogenic *Yersinia*. The *pst* gene on pPCP1, *caf1* on pMT1, and chromosomally located *ypo2088* have all been used as *Y. pestis* detection targets [11]. Another specific target gene (*pla*) encodes the pPCP1 plasmid-located plasminogen activator. These unique plasmids, however, are not present in all *Y. pestis* strains [12,13]. Despite *ypo2088* and *yih**N*, there are few other specific targets on chromosomes for the identification of *Y. pestis* [14,15]. Thus, although MALDI-TOF MS is increasingly used for *Yersinia*-species identification, it does not distinguish species that fall within the *Y. pseudotuberculosis* complex [16,17,18,19]. Although high-throughput sequencing-based whole-genome identification is the most accurate identification method, simple sequence alignments will still be prone to the types of errors mentioned above. Therefore, more reliable and accurate targets are needed to help researchers to quickly identify *Y. pestis* and distinguish it from its related species.

In this study, we electronically split the *Y. pestis* CO92 strain’s genomic sequence into millions of fragments comprising 100-bp windows in 1-bp steps. These fragments were used to compare and analyze the genome sequences from 539 strains of *Y. pestis* and 572 strains from other *Yersinia* species. From these, we identified 483 fragments containing *Y. pestis* species-specific single-nucleotide polymorphism (SNP) sites. Altogether, 97 *Y. pestis*-specific tags containing two or more SNP sites were screened out. These 97 tags are sufficient to efficiently distinguish *Y. pestis* from other closely related species. We used these 97 tags to write an automated program to help researchers quickly identify *Y. pestis* after they have obtained a draft genome sequence. We also designed a CRISPR-Cas12a-assisted SNP detection system using four of our newly developed *Y. pestis*-specific tags.

CRISPR-Cas systems are widely used for genome editing [20] and, in recent years, the trans-cleavage activity of Cas proteins has been discovered [21]. Detection methods based on CRISPR-Cas, which is now also considered a next-generation pathogen detection method, have become established [22,23]. DETECTR (DNA endonuclease targeted CRISPR trans-reporter) [24] and HOLMES (one-HOur Low-cost Multipurpose highly Efficient System) [25] can detect DNA sequences with attomolar sensitivity and high specificity using CRISPR-Cas12a [26]. Recombinase polymerase amplification (RPA) does not require thermal denaturation of the template and can be run at a low constant temperature. RPA has been successfully integrated into different detection platforms, from end-point lateral flow strips to real-time fluorescent detectors. The lateral flow dipstick (LFD) method is easy to use and its results easily interpretable. It can be used to detect proteins and nucleic acids. In this study, four tags selected from the aforementioned 97 *Y. pestis*-specific tags were designed for use with the Cas12a-based detection system in combination with RPA [27,28] and with LFD methodology [29,30] for easy visualization.

## 2. Materials and Methods

### 2.1. Strains Used in This Study 

*Y. pestis* 201 was isolated from *Microtus brandti* in Mongolia Inner, and is an avirulent strain in humans [31]. The *Y. pseudotuberculosis* 1682 (Yp1682) strain was isolated from a pig in Japan, the *Y. pseudotuberculosis* 1678 (Yp1678) strain was isolated from a person in Japan, and the *Y. pseudotuberculosis* 1688 (Yp1688) strain was isolated from a dog in Japan. They are part of the Beijing Institute of Microbiology and Epidemiology strain collection. The *Bacillus anthracis* A16PI2 strain is an attenuated strain with a deleted pXO2 plasmid [32]. *Staphylococcus aureus* ATCC49521 and *Escherichia coli* DH5α were used in the specificity tests. 

### 2.2. Acquisition of Genome Data for Y. pestis and Other Yersinia Species

We downloaded the genome data for *Y. pestis* and other *Yersinia* species strains from NCBI (https://www.ncbi.nlm.nih.gov/genome/?term=Yersinia, accessed on 12 February 2020). The *Y. pestis* CO92 strain was used as the reference. The DNA fragments on its chromosomal sequence (GenBank: NC_003143, 13-DEC-2020) were selected as the source of the specific tags. To obtain all the specific sites, we wrote a python script. On the *Y. pestis* CO92 chromosome, the DNA fragments that we conceptually cut into 100-bp windows in 1-bp steps were the candidate tags.

### 2.3. BLAST Analysis on DNA Tags and Screening for Y. pestis-Specific DNA Fragments

We used a local BLAST (BLAST-2.7.1+) to query the candidate tags in the genomes of other *Yersinia* species strains. When they appeared on the chromosomes of these strains (Query Cover = 100%; Percentage of Identity = 100%) they were removed. We then wrote a python script to query the remaining fragments on all the *Y. pestis* chromosomes, and to count the distributions of the fragment in each strain. The fragments distributed on all *Y. pestis* chromosomes were obtained, and they were considered to be unique DNA fragments belonging to *Y. pestis*. We wrote another python script to merge the obtained fragments into long non-overlapping DNA fragments so that all of the fragments would be specific for *Y. pestis*.

### 2.4. Constructing Specific Tags and Searching for SNPs for Detecting Y. pestis

We searched the *Y. pestis*-specific long DNA fragments against the chromosomes of other *Yersinia* species strains using local BLAST software (BLAST-2.7.1+). According to the BLAST results, we extracted the homologous fragments from each *Y. pestis*-specific long DNA fragment in other *Yersinia* species strains and made them into FASTA format files. These homologues FASTA DNA sequences were aligned with MEGA-X software to locate SNP sites distinguishing *Y. pestis* from other *Yersinia* species strains. After screening out the sequences containing two or more SNP sites, we were left with SNP-containing DNA fragments (100–200-bp long) for use as tags to specifically identify *Y. pestis*. To ensure the specificity of these tags, we performed an online BLAST program to search these tags (https://blast.ncbi.nlm.nih.gov/Blast.cgi, accessed on 25 February 2021) and remove those that appeared to be non-*Y. pestis*.

Based on this set of specific tags, we developed a mini software platform using python, based on the principle that if these tags appear in the genomes of the target strains, these strains will be identified as *Y. pestis*. We reasoned that the more tags that are identified, the more reliable the results will be. 

### 2.5. Establishing a CRISPR-Cas2a-Assisted Y. pestis Identification Method Based on SNPs

#### 2.5.1. Detecting SNP Sites in *Y. pestis* Using a PCR–Fluorescence Combined Method

This study used the CRISPR-Cas12a system for detection. The amplified product of the target fragment was added to the CRISPR-Cas12a system, and the signal sent by the single-stranded DNA probe was detected (Figure 1a). The Cas12a protein binds to the substrate DNA under the guidance of CRISPR RNA (crRNA) to form a Cas12a/crRNA/DNA complex. The complex can cleave single-stranded fluorescent DNA probes. When crRNA is consistent with the complementary sequence of the substrate DNA, it will emit strong fluorescence, but when the complementary sequence of crRNA and the substrate DNA are not completely consistent, weak fluorescence or no fluorescence is emitted (Figure 1d).

The sequences of the PCR primers used for detection are shown in Table 1. Formation of the Cas12a/crRNA/DNA complex requires a protospacer adjacent motif (PAM) (a TTTN sequence). When no PAM was present before the complementary sequence, we introduced it using PCR amplification primers (Figure 1b). To shorten the reaction time, the length of the DNA product was kept between 100–150-bp. The 17-base complementary sequence and the universal sequence (5’-AAUUUCUACUGUUGUAGAU-3’) formed a complete crRNA (Table 1) [33].

The complementary sequence of the crRNA and the target DNA together determine the efficiency of Cas12a protein cleavage. When the complementary sequence of the crRNA and the *Y. pestis* sequence completely match, strong fluorescence is emitted. Should the complementary sequence of the crRNA not completely match the target sequence in *Y. pseudotuberculosis*, weak fluorescence will be emitted for *Y. pseudotuberculosis*. The more mismatch sites occurring between the complementary sequence of the crRNA and the *Y. pseudotuberculosis* target, the weaker the fluorescence produced. Hence, the four tags containing two SNPs in the seven bases at the 5′ ends of the complementary sequence were selected (Table 1).

We next designed a 12-base, single-stranded DNA probe (5′-GAGACCGACCTG-3′. The base at the 5′ end of the fluorescent probe was labeled with HEX, and the 3′-end base was labeled with BHQ1. *Y. pestis* (*Y. pestis* 201) and *Y. pseudotuberculosis* (YP1682) genomes (10 ng/μL) were used as PCR amplification templates. The reagents used in the 50 μL reactions are shown in Appendix A. The PCRs comprised 5 min at 94 °C, followed by 30 cycles of denaturation at 94 °C for 30 s, annealing at 55 °C for 30 s, and extension at 72 °C for 30 s, with a final extension at 72 °C for 5 min. PCR amplicons (2 μL each) were added to the Cas12a reaction. The Cas12a reaction was conducted at 37 °C in a 20 μL volume (Appendix A). Fluorescence intensities were detected at 60 min (excitation wavelength 535 nm, detection wavelength 553 nm). The fluorescence intensity was detected on the Bio-Rad real-time PCR CFX96 instrument (Life Science, Hercules, CA, USA). The PCR amplifications used TaKaRa ExTaq™ Version 2.0 kit, and the primers were synthesized by Beijing Tianyi Huiyuan Biotechnology Co., Ltd. The LbCas12a (0.15 mg/mL) protein was expressed and purified as described previously [34]. The crRNAs and single-stranded DNA probes were synthesized by General Biosystems (Anhui) Co., Ltd., Chuzhou, China).

To evaluate the method’s specificity, we concurrently detected *Y. pestis* (*Y. pestis* 201), *Y. pseudotuberculosis* (Yp1682, Yp1678, and Yp1688), *Bacillus anthracis* (A16PI2), *S. aureus* (ATCC49521), and *E. coli* (DH5α). Genomic DNAs (50 ng/μL) from each bacterium were extracted as the templates. The fluorescence intensity detected every 12 min was statistically analyzed.

The sensitivity experiment involved diluting the *Y. pestis* 201 genome in DEPC-treated water. *Y. pestis* genomic DNA was used at different concentrations (10^6^, 10^5^, 10^4^, 10^3^, 10^2^, and 10^1^ fg/μL) to evaluate the method’s sensitivity. *Y. pseudotuberculosis* genomic DNA (10^6^ fg/μL) was used as the negative control. Assay sensitivity was calculated according to the fluorescence intensity emitted at 60 min post-reaction. 

#### 2.5.2. Detecting SNP Sites in *Y. pestis* Using a Combination of RPA and LFD

RPA amplification was performed using the TwistAmp^®^ Liquid exo kit (TwistDX, Maidenhead, UK). The GC content of the RPA primers (Table 1) was between 20% and 70%, and the Tm was between 50 °C and 100 °C. The reaction conditions were 37 °C for 40 min. The reagents volumes are shown in Appendix A. RPA combined with the LFD was then used to detect the four tags. The LFD probe contained 12 bases and its sequence was 5′-GAGACCGACCTG-3′. The base at the 5′ end of the probe was fluorescein-labeled, and the 3’-end base of the probe was biotin-labeled. The HybriDetect kit (TwistDX, Maidenhead, UK) was used for the strip-based detection. The Cas12a reaction product (10 μL) was added dropwise to the strip, the strip was placed at room temperature for 5 min, and 100 μL of buffer was added. After 2 min, the strip was removed from the buffer and the result was read immediately.

## 3. Results

### 3.1. Acquisition of Specific Fragments

We downloaded the genomes of 539 *Y. pestis* strains and 572 other *Yersinia* species strains (including 20 species of the *Yersinia* genus) from the NCBI genome database (Figure 2b). The *Y. pestis* CO92 strain’s chromosome was in silico cut into 4,829,756 fragments (100-bp windows in 1-bp steps) using a python script (Figure 2a).

### 3.2. Screening Results for the Y. pestis-Specific Fragments

Using a local BLAST (BLAST-2.7.1+), we removed the 4,191,017 fragments appearing in other *Yersinia* species strains, which left 638,739 remaining fragments. Surprisingly, when we analyzed their distributions on the *Y. pestis* chromosome, we found that the genome of one particular strain (Assembly ID: GCA_902387395.1, UHGG_MGYG-HGUT-02476) had a tag detection rate of only 1341 (0.21%) in the 638,739 fragments, whereas the tag detection rate of the other 538 *Y. pestis* strains was between 62.36% and 100%. After performing a whole genome cluster analysis, it was apparent that this strain was not *Y. pestis*, but *Y. pseudotuberculosis* (Figure 2d). Hence, we removed this strain from further analysis, which resulted in 35,680 fragments obtained for all 538 *Y. pestis* strains. These 35,680 tags were merged according to the overlaps between the fragments, and 833 specific fragments were obtained (Figure 2a). The length of these fragments was between 100-bp and ~ 411-bp.

### 3.3. Obtaining Y. pestis-Specific Tags

The 833 DNA fragments were searched for among other *Yersinia* species strains (572 strains) using a local BLAST software (BLAST-2.7.1+). The corresponding DNA sequences from each strain were extracted and 833 FASTA format files were constructed. Each of these files was aligned with MEGA-X software to find SNP sites that could distinguish *Y. pestis* from other *Yersinia* species or strains. As a result, we found 483 fragments containing SNP sites (Figure 2a). To improve the detection specificity, any sequences containing more than 2 (including 2) SNP sites were selected from the 483 fragments. Altogether, 97 fragments were located and used to specifically identify *Y. pestis* (Figure 2c). The 100–200-bp long sequences containing these SNP sites were used as *Y. pestis*-specific tags (Appendix A). To test the specificity of the 97 *Y. pestis*-specific tags on a larger database they were searched online (https://blast.ncbi.nlm.nih.gov/Blast.cgi, accessed on 25 February 2021). The results showed that the sequences of the *Y. pestis*-specific tags were completely consistent with the sequences of the *Y. pestis* species as a whole, and the Query Cover and Per. Ident (Percentage of Identity) values were 100%. The sequences of these *Y. pestis*-specific tags lacked complete consistency with non-*Y. pestis* species and strains and could therefore be used to identify *Y. pestis*.

We used these tags in combination to complete a convenient, practical automated program in python for users as an application to rapidly characterize *Y**. pestis* using nucleotide sequencing data. This python code is available at: https://github.com/844844/Identify_Y.pestis/tree/main (accessed on 24 March 2021). The program is the local version of Microsoft Windows, and strains can be one-click identified by inputting the sequencing data. The software makes it easy for users to understand the output analysis results, share data with colleagues, and make definitive judgments on the suspected isolates. 

### 3.4. Detecting SNP sites in Y. pestis by PCR-Combined Fluorescence Methodology

#### 3.4.1. CRISPR-Cas12a-Assisted *Y. pestis* Identification Using PCR-Combined Fluorescence Detection

We chose four *Y. pestis*-specific tags to detect *Y. pestis*, naming them YP-1, YP-2, YP-3, and YP-4 sites. We found that the fluorescence intensity produced by the detection of *Y. pestis* was much higher than that of *Y. pseudotuberculosis*, a statistically significant difference (Figure 3). The detection results for the four sites were the same, and the fluorescence intensity clearly distinguished *Y. pestis* from *Y. pseudotuberculosis*.

#### 3.4.2. Specificity of the Method Used to Differentiate *Y. pestis* from Other Bacteria 

In the previous experiment, we tested the assay’s specificity for *Y. pestis* and *Y. pseudotuberculosis* (Yp1682). *Y. pestis* and *Y. pseudotuberculosis* belong to the *Yersinia* genus, and their genomes are highly similar. To evaluate the specificity of the method, we concurrently detected two other *Y. pseudotuberculosis* (Yp1678, Yp1688), as well as *B. anthracis* (A16PI2; *Bacillus* genus), Gram-positive *S. aureus* (ATCC49521), and Gram-negative *E. coli* (DH5α). The results (Figure 4) showed that the fluorescence intensity for *Y. pestis* was significantly higher than that of the control species and strains. Thus, the four *Y. pestis*-specific tags displayed good specificity.

#### 3.4.3. Sensitivity of the Method Used to Differentiate *Y. pestis* from Other Bacteria 

We next evaluated the detection sensitivity of this method using *Y. pestis* genomic DNA as the template, and genomic DNA from *Y. pseudotuberculosis* (Yp1682) as the control. The results (Figure 5) showed that the minimum detectable concentration of YP-1, YP-2, and YP-3 sites was 10^3^ fg/μL, and the minimum detectable concentration of YP-4 was 10^6^ fg/μL.

### 3.5. Detecting SNP Sites in Y. pestis by RPA Combined with LFD

To shorten the detection time, we used RPA instead of the traditional PCR method to amplify the target DNA. The results in Figure 6a show that strong fluorescence was detected with *Y. pestis*, whereas only weak fluorescence was detected with *Y. pseudotuberculosis*. We next used RPA combined with LFD as the visualization method to identify *Y. pestis*. Figure 6b shows that the *Y. pestis* strain produced a visibly strong dark colored band on the test position of the strips, indicating that the test results were positive. In contrast, no bands in the test results area were visible for *Y. pseudotuberculosis*. These results indicate that the visualization method can distinguish *Y. pestis* from *Y. pseudotuberculosis*.

## 4. Discussion

The *Yersinia* genus is an *Enterobacteriaceae* family member. The genus currently includes three prominent human and animal pathogens: *Y. pestis*, *Y. enterocolitica*, and *Y. pseudotuberculosis* [35]. Of these, *Y. pestis* causes the fatal infectious disease called plague. The difficulty in distinguishing *Y. pestis* from *Y. pseudotuberculosis* and other closely related species is related to their high phenotypic and genetic similarities. The targets of PCR-based methods used for species identification are usually sequences on the unique pMT1 and pPCP1 plasmids, but using these plasmids can be ineffective in identifying atypical strains such as plasmid-deficient ones. There are a few reports about targets on chromosomes. Some targets are considered effective (e.g., *ypo2088* and *yihN*), while more are proven to lack specificity (e.g., *inv*, *entF3*, and *wzz*) [11]. Thus, here we have obtained 97 optional targets for detection on the *Y. pestis* chromosome, which provide valuable references for subsequent detection research.

In this study, we selected detection targets from chromosomal genome data in the NCBI database. These data comprised 539 strains of *Y. pestis* and 572 strains belonging to other closely related species. Using the *Y. pestis* CO92 strain as the standard in our experiments, we conceptually split its chromosomal sequence into millions of fragments to obtain individual 100-bp long fragments, each of which we called a “tag”. Using local BLAST software, the fragments present in the 572 strains with homology to *Y. pestis* were removed. The labels remaining in all *Y. pestis* strains (539 strains) were then screened out. During this process, we discovered that a strain of *Y. pseudotuberculosis* was mistakenly regarded as *Y. pestis* in the NCBI database. Through the fusion fragments, we obtained *Y. pestis*-specific fragments and 97 fragments containing more than 2 (including 2) SNP sites were selected as the selection tags. We envisage that the wide application of whole genome sequencing in public health fields will mean that our specific tags will become increasingly important. Our tags contain in-built multiple SNP sites, thereby ensuring that strains with poor sequencing quality are not missed during the detection process. The SNPs are not located on a virulence plasmid, and the *Y. pestis* strains (pMT1- and pPCP1-) were identified using chromosomal molecular markers, thereby providing sufficient information for bacterial trace analysis.

We then selected four of the tags to establish a highly specific detection method for *Y. pestis* based on CRISPR-Cas12a. The CRISPR-Cas12a system has been successfully used to detect a variety of pathogens, including SARS-CoV-2, the human immunodeficiency virus, and *Mycobacterium tuberculosis*, among others [36,37,38]. The CRISPR-Cas12a detection system is highly sensitive and can discriminate between SNPs [39]. Therefore, we employed the Cas12a protein and PCR combined with fluorescence methodology to detect the four *Y. pestis* tag sites. Our method can clearly distinguish *Y. pestis* from *Y. pseudotuberculosis* with high specificity at a minimum detection concentration of 10^3^–10^6^ fg/μL. When detecting SNP sites, a PCR amplification primer is needed to introduce the PAM sequence so that the primer position is fixed near to the detection site. When the PCR product’s length is 100–150-bp, the positions of the two primers cannot be freely selected, and this will result in the primers having different PCR amplification efficiencies and the minimum detectable concentrations will also differ. The limit of detection with tag YP-4 (Figure 5) was lower than that of the other tags but the specificity was acceptable. In theory, any SNP locus can distinguish *Y. pestis* from other closely related bacteria based on the current NCBI database. Two to three sites are sufficient for *Y. pestis* identification in practical applications. Thus, when the nucleic acid concentration is not extremely low, use of the Yp-4 site is also an option and its use was therefore not rejected in this study. To shorten the reaction time, we replaced PCR with RPA. The template amplification time was reduced from 90 min to 40 min, and the total detection time was shortened by 50 min. Because RPA does not require any equipment it is very suitable for on-site testing [40], and we used LFD instead of the fluorescence method to achieve visual detection of *Y. pestis*. 

This set of tags not only can be used for the in silico identification of genome sequences, but also can be employed as the targets of molecular biology methods for clinical and environmental sample identification. It is worth noting that we used the latest Cas12a protein in this study. Last, our set of tags can be used with new diagnostic technologies such as high-throughput DNA chips and biosensors, as well as with traditional quantitative PCR and other species identification methods. Plague outbreaks still occur from time to time, so early and unambiguous diagnostic detection is essential for panic avoidance and for successful treatment and disease prevention.

## 5. Conclusions

We obtained 97 electronic tags for highly-specific detection of *Y. pestis*. We also developed a method to visually detect *Y. pestis*, which can be used for the robust, specific, and portable detection of the *Y. pestis* genome. RPA-Cas12a-LFD has great potential for on-site *Y. pestis* detection.

## Figures and Tables

**Figure 1 pathogens-10-00562-f001:**
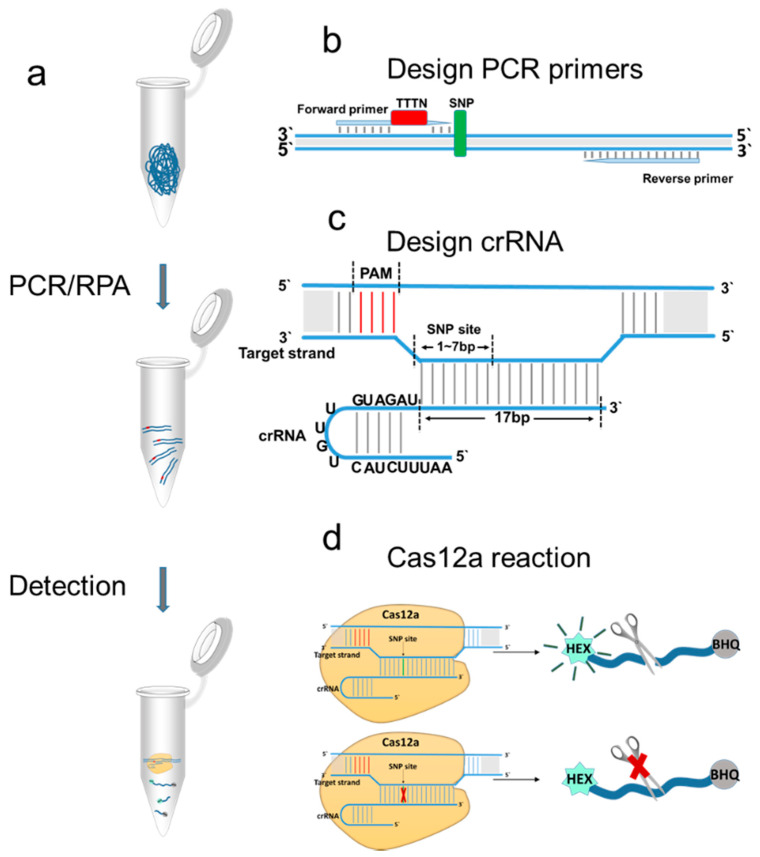
Experimental CRISPR-Cas12a detection-based design. (**a**) We first amplified the target fragment to improve the detection sensitivity. We then added the amplified substrate to the CRISPR-Cas12a reaction, and the signal from the probe was detected. (**b**) We used PCR primers to introduce the PAM sequence into the amplified product. (**c**) crRNA comprises a universal sequence and complementary sequence. (**d**) When crRNA and substrate DNA are completely matched, strong fluorescence is emitted. When crRNA and the substrate DNA are not completely matched, no fluorescence or weak fluorescence is emitted.

**Figure 2 pathogens-10-00562-f002:**
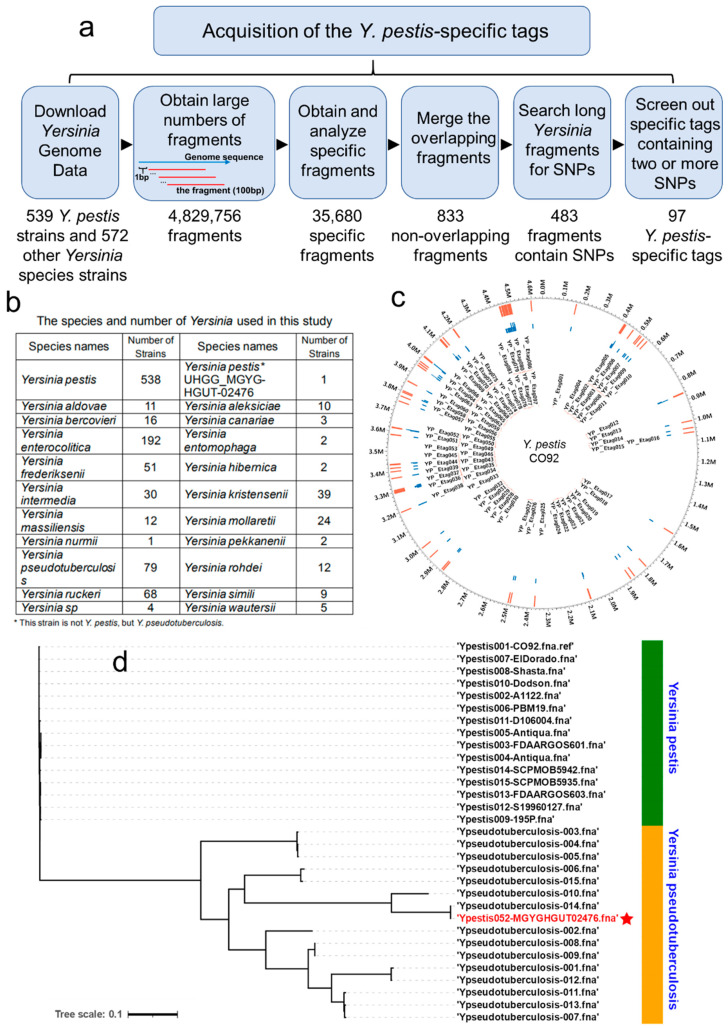
Acquisition of *Y. pestis*-specific tags. (**a**) The process used for screening and obtaining *Y. pestis*-specific tags. (**b**) The species and number of *Yersinia* genera used in this study. (**c**) Distribution of the 97 specific tags identified in *Y. pestis*. Outer circle to inner circle: the first red line in the ring shows the positions of the 97 tags on the chromosome; the second blue line in the ring shows the number of SNPs on the tags. (**d**) Cluster analysis of the uncertain *Y pestis* UHGG_MGYG-HGUT-02476 strain.

**Figure 3 pathogens-10-00562-f003:**
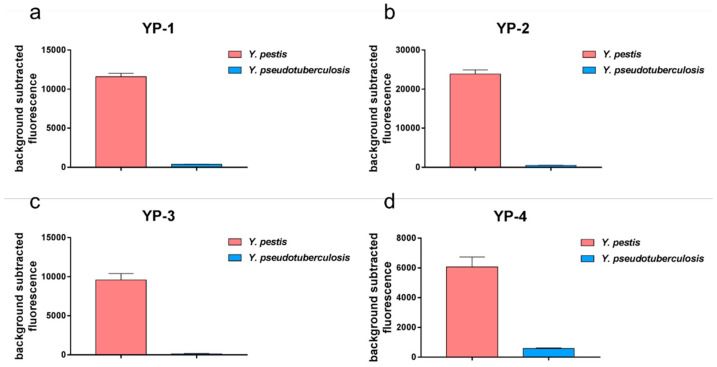
Detection of *Y. pestis* by PCR-combined fluorescence methodology. YP-1, YP-2, YP-3, and YP-4 tags were detected. When detecting *Y. pestis*, *Y. pseudotuberculosis* was used as a control. The fluorescence intensities emitted at 12 min post set-up were detected, and the data were judged. The fluorescence intensity detection of *Y. pestis* was higher than that of *Y. pseudotuberculosis* for the four tags. The results of detecting YP-1 tag (**a**), YP-2 tag (**b**), YP-3 tag (**c**) and YP-4 tag (**d**) from the two groups were analyzed using *t*-test, and the significant differences were noted (*p* < 0.01).

**Figure 4 pathogens-10-00562-f004:**
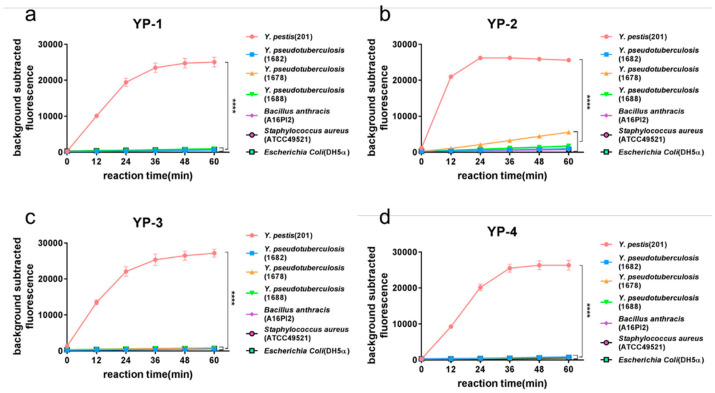
Specificity of the method used to differentiate *Y. pestis* from other bacteria. YP-1, YP-2, YP-3, and YP-4 tags were used to verify the method’s specificity. Seven strains including *Y. pestis* (*Y. pestis* 201), *Y. pseudotuberculosis* (Yp1682, Yp1678, and Yp1688), *B. anthracis* (A16PI2), *S. aureus* (ATCC49521), and *E. coli* (DH5α) were used to detect the four tags. Fluorescence intensity was detected at 0, 12, 24, 36, 48, and 60 min into the Cas12a reaction. The fluorescence intensity at each time point was used to plot and analyze the results. The results of detecting YP-1 tag (**a**), YP-2 tag (**b**), YP-3 tag (**c**) and YP-4 tag (**d**) showed that the tags differentiated *Y. pestis* from non-*Y. pestis* (two-way repeated measures ANOVA, *p* < 0.0001).

**Figure 5 pathogens-10-00562-f005:**
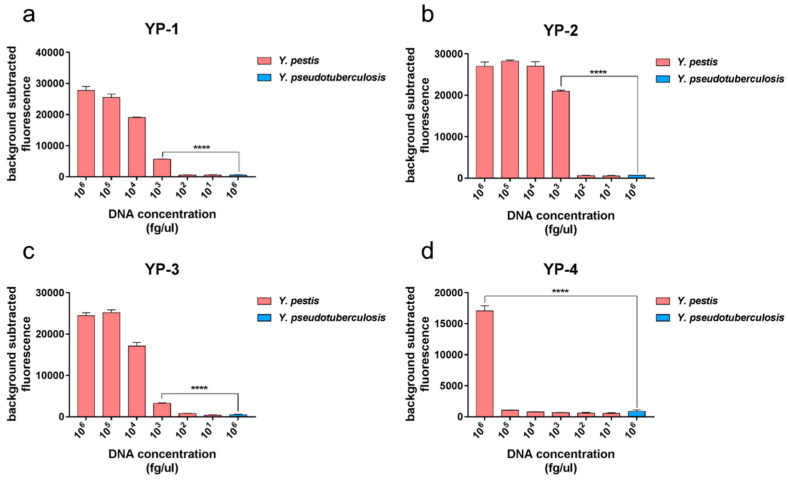
Sensitivity of the method used to differentiate *Y. pestis* from other bacteria. Bar charts show the fluorescence intensity detected at 60 min. *Y. pseudotuberculosis* was used as the control strain. The fluorescence values for YP-1 tag (**a**), YP-2 tag (**b**), and YP-3 tag (**c**) showed statistically significant differences compared with the negative control when the concentration reached 10^3^ fg/μL (*t*-test, *p* < 0.001). (**d**) The fluorescence values for the YP-4 tag showed statistically significant differences compared with the negative control when the concentrations of the sites reached 10^6^ fg/μL (*t*-test, *p* < 0.001).

**Figure 6 pathogens-10-00562-f006:**
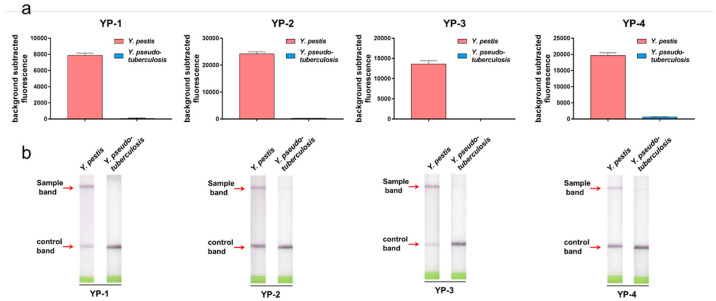
*Y. pestis* detection using the RPA method. (**a**) RPA combined with fluorescence was used to detect YP-1, YP-2, YP-3, and YP-4 tags. *Y. pseudotuberculosis* was used as the control strain for each tag. Fluorescence intensity was detected at a 12 min reaction time. The *t*-test method was used to analyze the results and a significant difference was observed between *Y. pestis* and *Y. pseudotuberculosis* for all the tagging sites (*p* < 0.01). (**b**) RPA combined with LFD to detect YP-1, YP-2, YP-3, and YP-4 tags. *Y. pseudotuberculosis* was used as the control strain for each tag. Fluorescence detection of YP-1, YP-2, YP-3, and YP-4 tags on LFD strips. A positive test band was visible for *Y. pestis* but not for *Y. pseudotuberculosis*.

**Table 1 pathogens-10-00562-t001:** Target sequences, primers, and crRNAs used for detection in this study.

Tag Names	Target Sequences	Primers		crRNA (5′-3′)
		Names	Sequences (5′-3′)	
YP-1	A*C*CAGACTCGCTCCACA	PCR-YP-1-F	AGGTGACAATTGTATACCTGCATAATTAATTAGCA*TTTA*	AAUUUCUACUGUUGUAGAUACCAGACUCGCUCCACA
		PCR-YP-1-R	ACAGATGTTGACTGGTGAGATGGTC	
		RPA-YP-1-F	GACAATTGTATACCTGCATAATTAATTAGCA*TTTA*	
		RPA-YP-1-R	ACAAATTTTACAGATGTTGACTGGTGAGATGGTC	
YP-2	C*CTCG*GTACTGTTGCCA	PCR-YP-2-F	CCGCCAGATCCACGCCC*TTTC*	AAUUUCUACUGUUGUAGAUCCUCGGUACUGUUGCCA
		PCR-YP-2-R	GCATCAACGGCATTTCGGCCA	
		RPA-YP-2-F	CCGGACAGATTGGCCGCCAGATCCACGCCC*TTTC*	
		RPA-YP-2-R	CAGCCGCATCAACGGCATTTCGGCCAATGGCAG	
YP-3	C*CT*ATGGCGTTCTCTAT	PCR-YP-3-F	TGCGAAATTGTACAAAAATCTCTGTTAGACT*TTTA*	AAUUUCUACUGUUGUAGAUCCUAUGGCGUUCUCUAU
		PCR-YP-3-R	CGCTGTTCTGCAACTTGAGTGACTAC	
		RPA-YP-3-F	GTGCGAAATTGTACAAAAATCTCTGTTAGACT*TTTA*	
		RPA-YP-3-R	GTAGTCACTCAAGTTGCAGAACAGCGTAAAACG	
YP-4	A*GA*GACAAATATCACCA	PCR-YP-4-F	TTGGCTTCAAGCGATTCCAGTCAA*TTTA*	AAUUUCUACUGUUGUAGAUAGAGACAAAUAUCACCA
		PCR-YP-4-R	GCATGGAGCTATTATGACAAGAACCGG	
		RPA-YP-4-F	AATCGTTGGCTTCAAGCGATTCCAGTCAA*TTTA*	
		RPA-YP-4-R	TAGAAGCATGGAGCTATTATGACAAGAACCGG	

* SNP sites used to distinguish *Y. pestis* from non-*Y. pestis*. Italic text indicates PAM sequences. Underlined parts indicate the positions of the complementary sequences.

## Data Availability

Not Applicable.

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
