# Peer review of "Obtaining Specific Sequence Tags for Yersinia pestis and Visually Detecting Them Using the CRISPR-Cas12a System"

_pathogens, 2021, doi:10.3390/pathogens10050562_

Round 1

Reviewer 1 Report

Readers are provided with a not bad description of an elaborate set of studies aimed at developing a reliable methodology for identifying the infamous plague pathogen, which in historical times alone claimed several hundred million human lives.

The authors have achieved the desired result. Particularly impressive is the combination of high sensitivity and specificity with a short-term identification process, ensuring the timely initiation of antimicrobial therapy.

It seems to me that English should be minor corrections.

Reviewer 2 Report

The manuscript describes the selection of chromosomal Y. pestis-specific tags by analyzing sequences of other Yersinia species. Then, four tags were successfully used to design a Cas12a-based detection system, which can be useful for plasmid-independent diagnostic of plague pathogen. The manuscript is well-written, and conclusions correspond to the obtained data.  The reviewer has no major concerns on the manuscript.  

The minor comments are the following:

p.1, line 34. The first plague pandemic (Justinian plague) happened in sixth century, not in the third century.

p.2., line 74, typo ypo2088.

p.2, lines 74-76 and page 11, line 346. There is another chromosomal target widely used for plague diagnostic, gene yihN, see Stewart e.al, PMID: 18287295. This should be mentioned and referred.

p.3, line 123. Figure 2a referred before Figure 1, need to redo Figure numbers.

p.3, lines 136 & 140.  Replace here and further in the text, as well as in Fig. 2a, “Y. pestis homologous strains” to “other Yersinia species strains”

p.4, lines 152-154. Indicate source CRISPR-Cas12a system, if commercial, provide corresponding information here or in p.5, line 187.  

p.12, lines 388-390. Check grammar in this sentence, re-write.

Table S4. Provide the name of the strain and exact accession number used to define start and end sites for the tags.
